# Characteristics of Arsenic Leached from Sediments: Agricultural Implications of Abandoned Mines

**Soonho Hwang** [1] , **Younggu Her** [2] , **Sang Min Jun** [1], **Jung-Hun Song** [2], **Goontaek Lee** [3] **and Moonseong Kang** [4,*]

1 Department of Landscape Architecture and Rural Systems Engineering, Seoul National University, Seoul 08826, Korea; ynsgh@snu.ac.kr (S.H.); luckysm1@snu.ac.kr (S.M.J.)
2 Department of Agricultural and Biological Engineering/Tropical Research and Education Center, University of Florida, Homestead, FL 33031, USA; yher@ufl.edu (Y.H.); junghunsong@ufl.edu (J.-H.S.)
3 National Instrumentation Center for Environmental Management, Seoul National University, Seoul 08826, Korea; gtlee@snu.ac.kr
4 Department of Landscape Architecture and Rural Systems Engineering, South Korea Department of Rural Systems Engineering, Research Institute for Agriculture and Life Sciences, Institute of Green Bio Science and Technology, Seoul National University, Seoul 08826, Korea
* Correspondence: mskang@snu.ac.kr; Tel.: +82-2-880-4582

**Abstract:** Heavy metals, including arsenic from abandoned mines, are easily transported with sediment and deposited in waterbodies such as reservoirs and lakes, creating critical water quality issues when they are released. Understanding the leaching of heavy metals is necessary for developing efficient water quality improvement plans. This study investigated how arsenic leaches from different soil and sediment types and responds to hydrologic conditions to identify areas susceptible to arsenic contamination. In this study, batch- and column-leaching tests and sequential extraction procedures were used to examine arsenic leaching processes in detail. The results showed that most arsenic-loaded sediments accumulated in the vicinity of a reservoir inlet, and arsenic in reservoir beds have a higher leaching potential than those from agricultural land and stream beds. Arsenic deposited at the bottom of reservoirs had higher mobility than that in the other soils and sediments, and arsenic leaching was closely associated with the acidity of water. In addition, arsenic leaching was found to be responsive to seasons (wet or dry) as its mobilization is controlled by organic compounds that vary over time. The results suggested that temporal variations in the hydrochemical composition of reservoir water should be considered when defining a management plan for reservoir water quality.

**Keywords:** arsenic; leaching; sediment; heavy metal; reservoir; abandoned mine

## 1. Introduction

Acid mine drainage and materials left at abandoned mines can cause significant pollution problems [1–5]. Sediments loaded with heavy metals, which have deleterious effects on human, animal, and plant health, can be transported by runoff to downstream waterbodies such as streams, reservoirs, aquifers, and estuaries [6–9]. In particular, rice is known as a major source of arsenic exposure that could lead to critical human health issues, especially in south and southeast Asian countries [10–12]. If the sediments of a small agricultural reservoir for irrigation are contaminated with arsenic, leached arsenic from the transported sediment could cause critical water quality issues and food security. Therefore, for efficient management of the quality of irrigation water, understanding of how heavy metals leach from contaminated sediments and soils is required [13].

Many studies have investigated quality assessment of many ions to impact on the irrigation water, including sodium (Na), potassium (K), calcium (Ca), magnesium (Mg), bicarbonate ($HCO_3$),

chloride (Cl), and sulfate (SO$_4$) [14–16], as well as heavy metal leaching from various media, including sewage sludge [6,17,18], industrial waste [19], coral ash [20–22], tropical soils [23], and mine tailings [24,25]. However, the previous research has mainly considered the waterbodies of streams and rivers or relatively large-scale lakes and reservoirs [22,23,26], and the leaching of arsenic from the bottom of an irrigation reservoir has not been a focus of previous studies even though it has significant implications for agriculture. Small agricultural reservoirs have a relatively small storage capacity, and the water level changes frequently depending on the hydrological process with rainfall related watershed runoff and irrigation. Hence, the sediments of the reservoir are exposed to various environmental conditions, e.g., such as air during drought periods, and become submerged in water when high water levels are present. As arsenic leaching processes are sensitive to the physical and chemical conditions to which they are exposed, it is important to have a detailed understanding of how the processes respond to various environments under these water conditions. In addition, as various studies stressed and analyzed deposition [27] and diffusive gradient [28–31] characteristics related to the arsenic bioavailability (heavy metals and other persistent pollutants) in water and sediments, we considered the seasonal effects (differences of the water quality between wet season and dry season, especially soluble organic matter) on arsenic leaching characteristics.

In this study, we examined the process by which arsenic leaches from reservoir sediments with the goal of providing data required to develop an arsenic management plan for reservoirs and demonstrating the implications of arsenic-contaminated sediments on the quality of water in an agricultural reservoir. In this study, we compared the characteristics of soil and sediment samples which were taken at a paddy field and in the beds of a downstream stream and reservoir in an agricultural watershed. Multiple tests, including tests and sequential extractions, were conducted in a detailed investigation of the arsenic leaching process. In addition, to understand the long-term impacts of arsenic leaching on water quality, column-leaching tests were also applied to sediments deposited in the reservoir bed. In the column-leaching test, water quality in the wet and dry seasons were separately considered to understand the effects of different hydrologic conditions on arsenic leaching and contamination in the reservoir.

## 2. Materials and Methods

### 2.1. Study Area and Soil Preparation

This study focused on a watershed in the mid-western region of South Korea, where there are several abandoned gold mines. The watershed is mainly covered by forests (82.4%) and agricultural land (7.9%), and water moves from an upstream area of 7.4 km$^2$ to a downstream agricultural reservoir through two main streams (Figure 1). The streams are subject to flash floods as the upstream drainage areas are steep.

Gold mines scattered within this watershed were closed approximately 50 years ago and two of them were identified only recently (Figure 1). The mines had not been maintained appropriately and their tailings disposals are suspected to be the source of arsenic contamination across the watershed. Heavy metals become attached to soil particles and are transported with sediment to downstream overland areas, streams, and reservoirs in runoff from storm events. Sediment and heavy metal loads are then deposited on the reservoir bed. Under certain appropriate conditions, arsenic may leach out of the sediment and cause water quality and health issues such as arsenic poisoning. To understand the condition-dependent characteristics of arsenic leaching processes, soil samples were taken from upland agricultural fields and the beds of the downstream reservoir and streams. Reservoir water was sampled on 23 August 2017 (wet season) and 27 September 2017 (dry season) to track temporal changes of arsenic loads. A total of 18 water samples were collected from multiple sampling points (six points) and depths (three depths) in the reservoir.

This study used the following three different tests to investigate the arsenic leaching processes and their responses to the external conditions in detail: the batch- and column-leaching tests and

sequential extraction. For the batch-leaching test, the soil samples were collected from the reservoir bed. To identify critical arsenic sources among the various soils, agricultural areas and stream beds were also analyzed using the batch-leaching test. Soil samples for the column-leaching tests were prepared by mixing samples taken from six locations on the reservoir bed and then air-dried and sieved at 2 mm. In this study, the same method was applied to soils and sediments for measuring the arsenic concentration.

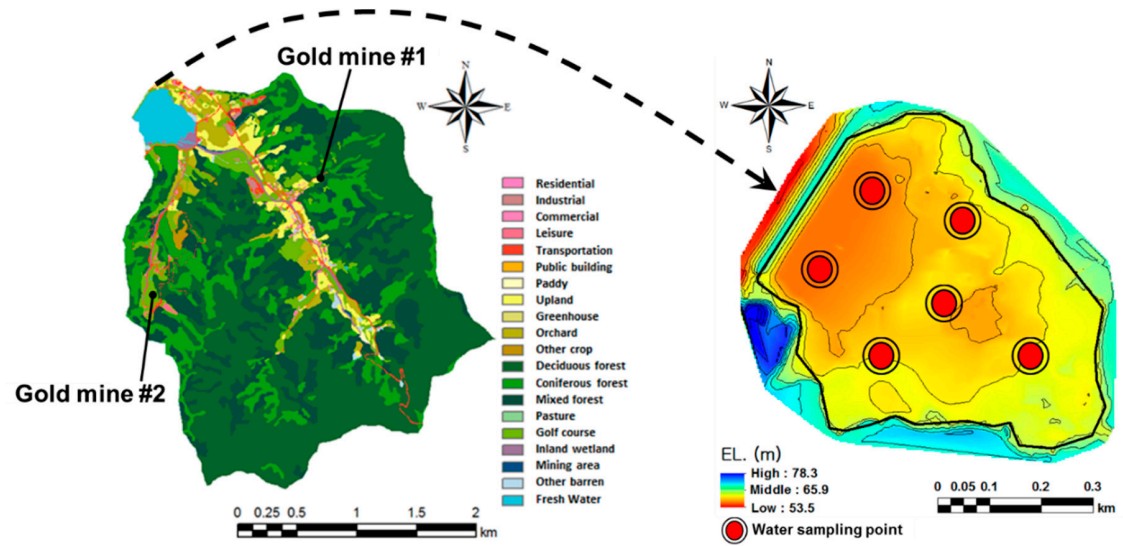

**Figure 1.** Study areas and sampling points for arsenic-laden sediment.

## 2.2. Batch-Leaching Test by Toxicity Characteristic Leaching Procedure (TCLP)

The toxicity characteristic leaching procedure (TCLP) has been used widely to analyze the characteristics of arsenic leaching in solidified contaminated soils [32,33]. However, the TCLP requires a pretest because different extraction solvents can be applied depending on the acidity levels (pH) of the samples. To prepare a soil–water mixture, 96.5 mL of water was added to a 5 g soil sample (particle sizes less than or equal to 1 mm) in a 500 mL beaker. The soil–water mixture was then stirred in a watch glass dish for 5 min and its pH was measured. If the pH became less than 5.0, 5.7 mL of glacial acetic acid ($CH_3CH_2OOH$) with a pH of $4.93 \pm 0.05$ was added to the mixture, which was then kept aside for 10 min so that any additional reactions could occur. If the pH became greater than 5.0, 3.5 mL of 1 N HCl (hydrochloric acid) was added instead of the $CH_3CH_2OOH$. In this study, the pH of all samples became less than 5.0 after the addition of the 1 N HCl. Once an extraction solvent had been determined, soil samples of more than 100 g were mixed with extraction fluid ($CH_3CH_2OOH$) in the ratio of 1:20, and the mixed samples were placed in an incubator shaker set to $30 \pm 2$ rpm and a temperature of $23 \pm 2$ °C. The mixture was left shaking for $18 \pm 2$ h and then filtered with glass fiber filter papers (pore size: 0.6 to 0.8 μm). Finally, the concentration of the leached heavy metals was measured using inductively coupled plasma atomic emission spectroscopy. Each of the tests was repeated nine times (repeated three times for each of three soil samples) in this study.

## 2.3. Sequential Extraction Procedure

A sequential extraction procedure was used in this study to determine the phase distribution and mobility of arsenic in the sampled soils. Sequential extraction procedures have the advantage of providing data on the specific forms of each metal and their behavior under various environmental conditions [34]. In addition, the procedures can provide the data required to assess the mobility and bioavailability of heavy metals in soils [7,35]. In this study, the sequential extraction method was used to examine the arsenic leaching processes occurring in soils sampled at agricultural fields and in the

beds of a downstream stream and reservoir. The overall procedures for the sequential extraction are described in Table 1; more details are given in [36], where the steps used in this study were taken from.

**Table 1.** Sequential extraction procedure for the fractionation of arsenic (adapted from [36]).

| Form | Phase | Extractant |
|---|---|---|
| 1 | Nonspecifically sorbed | $(NH_4)SO_4$ |
| 2 | Specifically sorbed | $(NH_4)H_2PO_4$ |
| 3 | Amorphous and poorly crystalline hydrous oxides of Fe and Al phase | $NH_4$ oxalate buffer (pH 3.25) |
| 4 | Well-crystallized hydrous oxides of Fe and Al | $NH_4$ oxalate buffer + ascorbic acid (pH 3.25) |
| 5 | Residual | Residual ($HNO_3$, $H_2O_2$) |

*2.4. Column-Leaching Test*

Reservoir water moves slowly and is often stratified vertically by density and temperature gradients. Thus, a reservoir would not be quickly contaminated with heavy metals that leach from sediment. Leached heavy metals can react with various other environmental processes (e.g., temperature variations and redox conditions), and their concentrations and leaching rates can increase over the years. A column-leaching test was carried out using samples of water from different conditions (deionized water, water from the wet season, and water from the dry season) to examine arsenic leaching from sediment that had been on the reservoir bed for a long time and the responses of the leaching processes to hydrological changes in the reservoir.

Sediment collected from the bed of the study reservoir was completely mixed and poured into a series of columns (diameter = 10 cm; height = 44 cm). The columns were fitted with an up-flow system to simulate the movement of groundwater toward the reservoir bed in the hyporheic zone. The sediment from the reservoir bed was sampled every day for 15 days. A homogenized sediment sample was prepared by mixing 30 kg of sediment for 24 h (using a V-mixer). The column-leaching test was performed under three conditions using the deionized water (control group), the wet season water (wet season group), and the dry season water (dry season group), respectively. The details of the column test are provided in Figure 2 and Table 2.

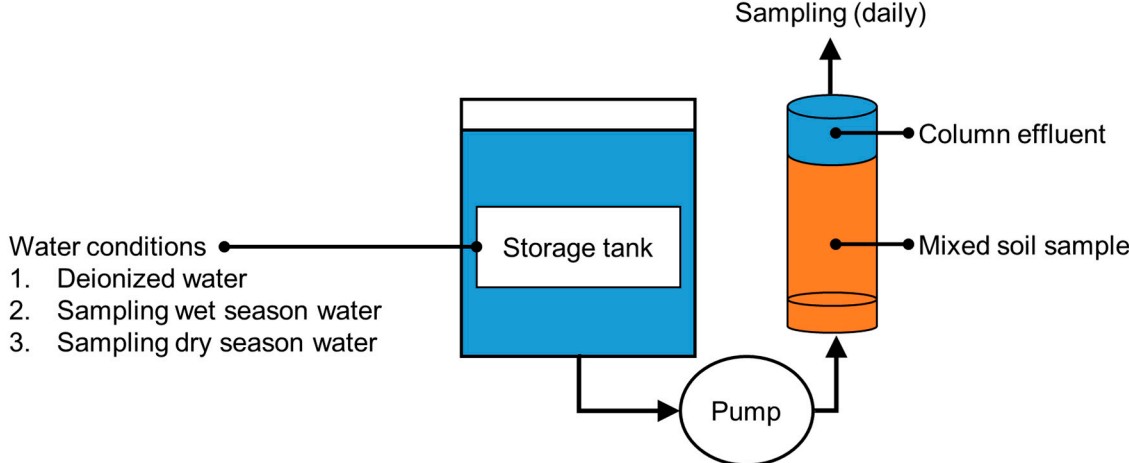

**Figure 2.** Schematic diagram of the column-leaching test (adapted from [37]).

The physicochemical properties of the soil samples used in the column-leaching test are shown in Table 3. The pH of the soil samples was slightly acidic (6.1) and the cation exchange capacity (CEC) was less than those found in other studies [6,36,38,39]. The soil test showed that the soil is loamy sand; its arsenic concentration was 56.34 mg/kg.

**Table 2.** Details of the column-leaching test.

| Category | Details |
|---|---|
| Column specifications and settings | Diameter = 10 cm, height = 44 cm; filled with mixed soil (grain size <2 mm) up to a height of 20 cm (total height = 44 cm); flow rate = 0.91 mL/min and 2 pore volumes (PVs) daily. |
| Sampling | Duration of experiment = 15 days; sampling effluent rate = daily. |
| Soil and water usage | Soil = 2438.4 g/column; amount of water used = 0.658 L/PV × 2 PV/day × 15 days = 19.74 L. |

**Table 3.** Physicochemical properties of soil samples used in column-leaching test.

| pH | Organic Matter (%) | Cation Exchange Capacity (CEC) (cmol/kg) | Total Nitrogen (TN) (mg/kg) | Total Phosphorus (TP) (mg/kg) | Available Phosphorus (mg/kg) | As (mg/kg) | Texture |
|---|---|---|---|---|---|---|---|
| 6.1 | 2.68 | 6.19 | 1120 | 374.2 | 21.9 | 56.34 | Loamy sand |

The physicochemical properties of water samples taken in the wet and dry seasons and studied using the column-leaching test are shown in Figure 3. The total phosphorus (TP) and total organic carbon (TOC) concentrations of wet season water were lower than those of the dry season water, but the total nitrogen (TN) of water sampled in the wet season was higher than that of the dry season. The arsenic concentrations in the wet season were slightly higher when dry season water was tested.

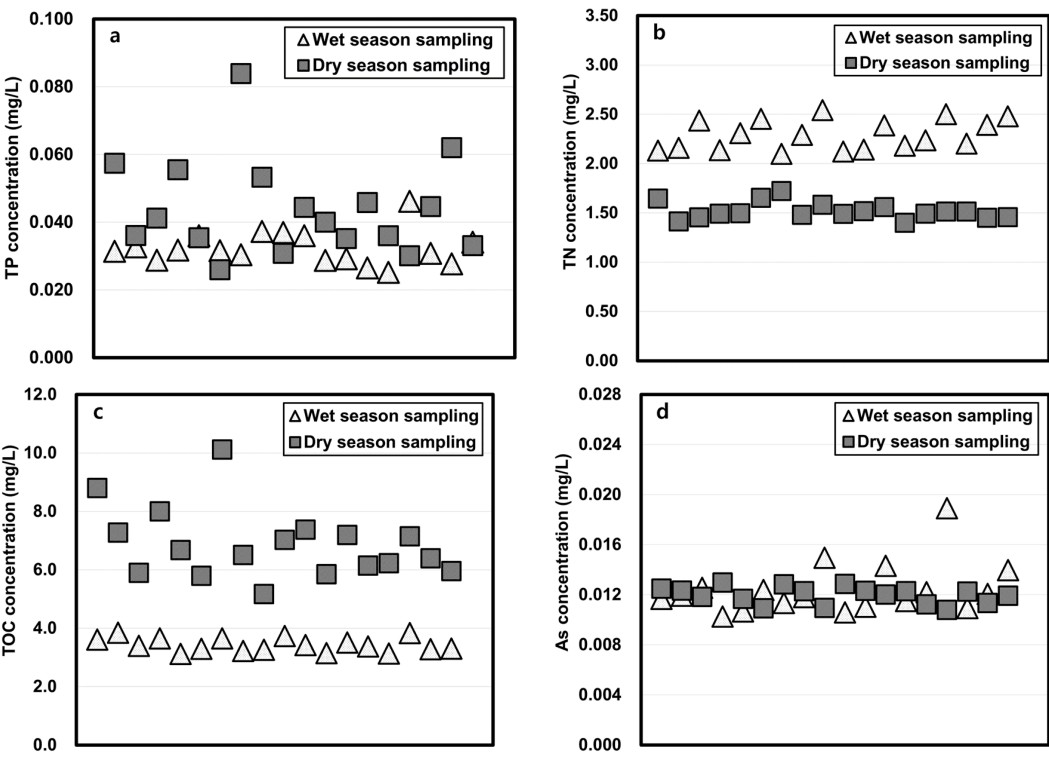

**Figure 3.** Physicochemical properties of reservoir water in wet and dry seasons. (**a**: TP, **b**: TN, **c**: TOC, **d**: As)

In the test, the total (accumulated) amount of arsenic leached was calculated using the following relationship:

$$\text{Total amount leached (mg)} = \text{Arsenic concentration (mg/L)} \times 0.658 \text{ L/PV} \times 2 \text{ PV/day} \qquad (1)$$

where L is liters and PV denotes pore volume.

## 3. Results

### 3.1. Batch-Leaching Test Results

The arsenic concentrations of soils and sediments sampled at three different locations were analyzed using the TCLP test (Figure 4). The highest average arsenic concentration was found in effluent from the reservoir sediment (0.026 mg/L); the average concentrations coming from the agricultural land soil (0.018 mg/L) and the stream sediment (0.019 mg/L) were similar to each other, but the concentrations of the soil samples from agricultural land show considerably greater variability (Figure 4). This means that the arsenic concentrations in each agricultural land of this study area varied, and were not considered to be highly contaminated; however certain agricultural lands which were readjusted using soil highly contaminated with mine tailings when the arable lands were altered show a high level of arsenic contamination.

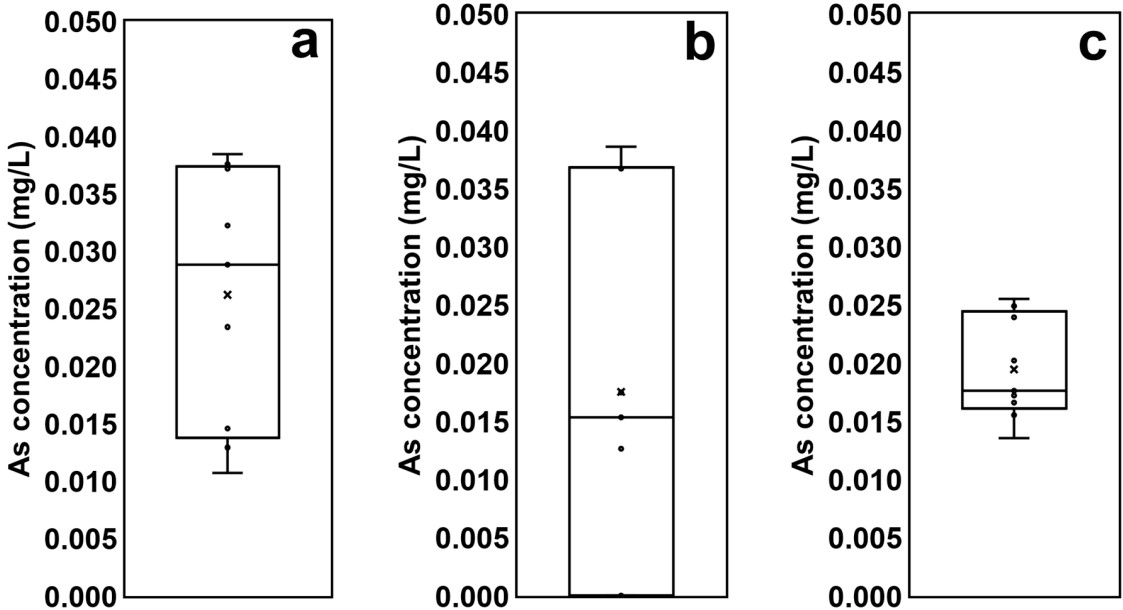

**Figure 4.** Comparison of arsenic concentrations derived from the batch-leaching test: (**a**) the reservoir sediment; (**b**) the agricultural land soil; (**c**) the stream sediment.

The contamination potential of soils and sediments is explained by the mobility of pollutants including arsenic. Arsenate ($AsO_4^{3-}$) and arsenite ($AsO_3^{3-}$), anionic arsenic compounds, form chelates and precipitate in combination with multiple cationic metals. When arsenic adheres to soil and sediment particles with metallic substances (e.g., steel and aluminum), the iron/arsenic (Fe/As) ratios increase and the mobility of arsenate may decrease. In this study, we investigated the Fe/As ratios of the soil and sediment samples as an indicator of their arsenic mobility. The average Fe/As ratio of the reservoir sediment (368) was lower than those from the other soils and sediments (1819 for the agricultural land soil and 1914 for the stream sediment), indicating that As in the reservoir sediment was more mobile and had a higher leaching potential than that in other soil and sediment.

### 3.2. Sequential Extraction Procedure

The As in the reservoir sediment was predominantly associated with amorphous and poorly crystalline Fe and Al hydrous oxides (average percentage = 38.86%), but there was a relatively low average concentration of As found in soil from the agricultural land and the stream bed samples (19.42% for the agricultural land and 13.41% for the stream bed; Figure 5). The main extraction percentages of As in soil from the agricultural land and stream bed were 36.34% and 45.21%, respectively, within well-crystallized Fe and Al hydrous oxides. There was a relatively lower percentage of As in the

residual fraction in the reservoir sediment than in samples from the agricultural land and stream bed sediment. This result suggests that As found in the reservoir bed sediment was more easily mobilized than that found in the other locations.

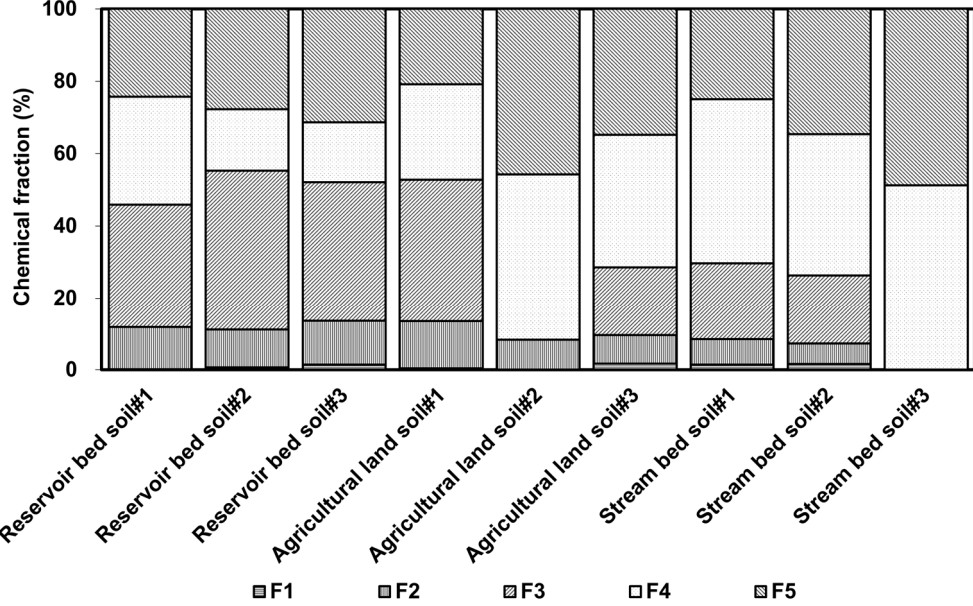

**Figure 5.** Distribution ratio of arsenic in soil as determined by continuous extraction ("F" means form in Table 2).

## 3.3. Column-Leaching Test Results

The concentration of leached As of the control group increased substantially (from 0.13 to 0.56 mg/L) in the first seven days, but then stabilized (between 0.57 and 0.6 mg/L) for the remainder of the 15-day test period (Figure 6). In the case of the wet and dry groups, the As concentrations also increased considerably during the initial stage of the tests, as was observed in the control group. However, unlike that group, they started decreasing during the middle of the test and reached similar values that were much lower than the control by the end of the test (Figure 6).

Such a result could be explained by the difference between the acidity and temporal variations of the three groups (Figures 6 and 7). As mobility and concentration tend to increase with increases in pH [39]. As mobility is very sensitive to acidity under both oxidizing and reducing conditions [39,40]. In the column-leaching test, pH values were used as an indicator of acidity (Figure 7). The pH values of the effluent ranged from 7.8 to 8.3 for the control group. Water samples taken in the wet and dry seasons had similar pH values, although they were a bit lower than the control, ranging from 7.5 to 8.0. The pH of the effluent increased gradually for all groups by the fifth day of the test. From the fifth to the tenth day, the pH of the control group continued to increase but the acidity of the other two groups (wet season and dry season water samples) remained in the neutral range. After the tenth day, the acidity of the all groups decreased gradually. Overall, the acidity levels were highly correlated to As concentration, and the test showed that the control group was relatively more alkaline than the others.

Another important point of the result in Figures 6 and 7 was that the samples of the wet and dry season group showed a similar temporal variation in the pH pattern, but those of As concentrations started diverging on the fifth day (Figure 6). The result can be explained by the water quality used in the experiments. The wet season water had higher TN concentrations, but lower TP and TOC concentrations than the dry season water (Figure 3). Since soluble organic compounds can act as chelators [39], the wet season water had higher arsenic leaching concentrations after the fifth day because of the reduced chelation of arsenic occurring with this water.

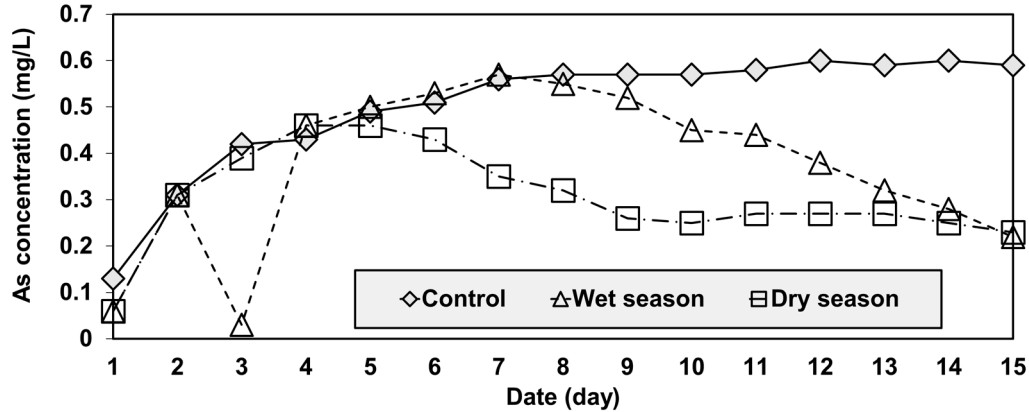

**Figure 6.** Temporal variations of arsenic concentration in the reservoir.

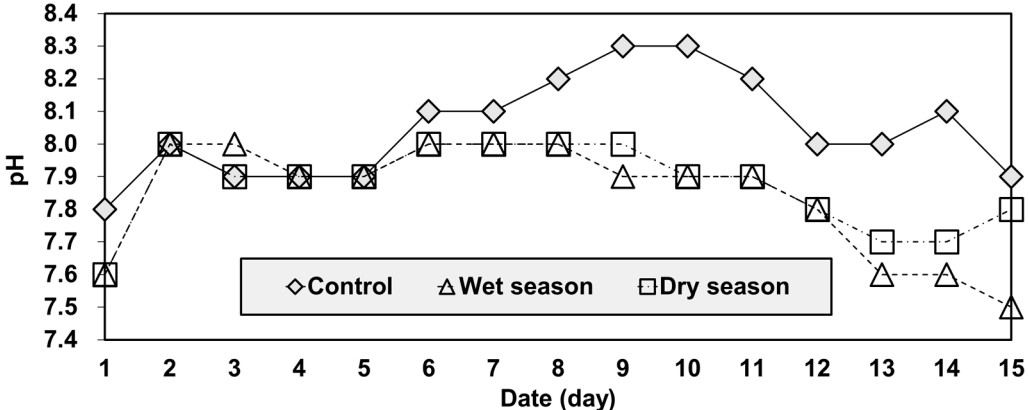

**Figure 7.** Temporal variations of pH in the reservoir.

At the end of the test period, the final cumulative amounts of arsenic leached were 7.52 mg for the control group, 5.62 mg for the wet group, and 4.58 mg for the dry group (Figure 8). The control and dry season water showed the largest and smallest cumulative quantities of leached As, respectively. The difference between the quantities of As leached in the wet and dry season water samples was about 20%, suggesting that reservoir water should be more carefully investigated during a wet season.

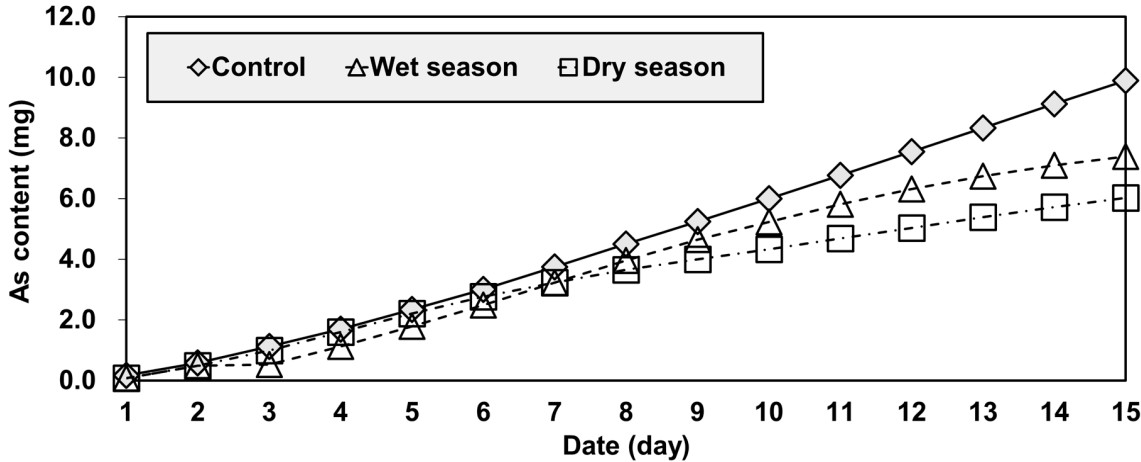

**Figure 8.** Comparison of the cumulative amount of arsenic leached from the three samples.

## 4. Conclusions

In this study, we found that As attached to sediment deposited on a reservoir bed had a relatively high leaching potential compared to that found in agricultural land and stream beds. Such a result was attributed to the fact that the reservoir sediment had high nonspecifically and specifically sorbed As fractions, and a corresponding low residual fraction. The As concentrations vary with the pH and redox conditions of the ambient water. Although the water samples showed a similar temporal variation pattern in their acidity over time in the column-leaching test, water sampled in the wet season was found to have higher As concentrations than the other samples, and to have leached a higher cumulative amount of the same to the ambient water due to differences in the amounts of organic compounds the samples contained. This suggested that seasonal variations in geochemical processes should be considered in planning for water quality management. It was also found that more As was leached out of sediment when the concentrations of organic compounds in reservoir water are low and when the water has a low acidity (i.e., a high alkaline pH), implying that it is necessary to closely monitor the physicochemical state and variations of As concentrations in reservoir water to ensure acceptable quality of this source of irrigation water.

Finally, to help farming activity and supply irrigation water safe from arsenic contamination, the local government and irrigation water management agency need to regularly monitor water quality changes of the reservoir and stream. For effective reservoir water quality management practices, the local government should include a plan for the removal of As-contaminated sediment from reservoir beds of interest and consider ways to screen As transported with sediment into reservoirs. Another important point is that because there was a large variability in the arsenic leaching concentrations of the samples from agricultural land, the local government and farmers need to survey the contamination level of most agricultural lands and restore the contaminated soil immediately.

**Author Contributions:** Conceptualization, S.H., Y.H. and M.K.; methodology, G.L.; investigation, S.H., Y.H., S.M.J. and J.-H.S.; writing—original draft preparation, S.H.; writing—review and editing, Y.H. and M.K.

**Funding:** This research was supported by "Establishment of water quality improvement measures in Ipjang reservoir" project which was funded by Korea Rural Community Corporation.

**Acknowledgments:** We thank the reviewers for their constructive comments and suggestions on this article. We also thank Cho for his help in conducting this project.

**Conflicts of Interest:** The authors declare no conflict of interest.

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
