# Peer review of "Characteristics of Arsenic Leached from Sediments: Agricultural Implications of Abandoned Mines"

_applsci, doi:10.3390/app9214628_

Round 1
Reviewer 1 Report
Despite its evident interest, the paper left me with a mixed impression.
Pro:
- its focus on As, which fits the objective of assessing mining impacts,
- a not so frequent investigation of sediment sinks in the downstream drainage of mines - a frequently overlooked issue - for this reason at least, the paper is more than welcome
- the careful analysis of potential As release through sequential leaching tests
Con:
- only As release to water is taken into account, not the potential sedimentation, fixation or trapping, especially in relation with redox maturation in reservoirs. This is a major flaw of the research design, even it is acceptable to focus on the sole release phenomenon. But the competing phenomena need to be identified, even through references,
- no taking into account, for potential downstream impacts, of other As carriers - especially solutes or particles "flying" over reservoirs, or sorption/desorption phenomena in the river bed downstream (sediment/water exchange),
- recommendations for reservoir sediment removal do not take into account the possibility of increased impacts through remobilisation - and do not compare with reservoir bed confinement. As expressed in the conclusions, they may be severely misleading for the design of a remediation operation.
This needs some rewriting/rethinking of the discussion and conclusions.
Not many issues in the results section, apart from being too focused. I noticed:
- Line 173: "the concentrations coming from the agricultural land (0.018 mg/L) and the stream bed (0.019 mg/L) were similar to each other" True for stream sediments, but only on average for soils, this hides their very large variability (Figure 3).
- Line 181 "soil from the stream bed" do you mean stream sediment ? or soil formed in the stream bed ?
Some very minor formal corrections:
Arsenate (AsO43) and arsenite (AsO33) use exponent case for ionic charge
This paper should make it to publication but needs a revised perspective.
Good luck !
Author Response
Responses to Reviewer’s Comments for the Manuscript, applsci-617476
Characteristics of Arsenic Leached from Sediments: Agricultural Implications of Abandoned Mines
Reviewer
Comment R.1:
- only As release to water is taken into account, not the potential sedimentation, fixation or trapping, especially in relation with redox maturation in reservoirs. This is a major flaw of the research design, even it is acceptable to focus on the sole release phenomenon. But the competing phenomena need to be identified, even through references,
Response to Comment R.1: Thank you for your constructive and positive comments on our study and this manuscript. As you point out, the factors are the very important independent variables and we have to consider and analyze the many factors to influence the leaching mechanisms from the reservoir sediment. However, in this research, we just wanted to analyze the differences the leaching mechanisms by the water conditions and although we didn’t show the all the water quality monitoring results on this manuscript, the SS concentration of two different season water show similar concentrations. So, we guess and derive the factor the As bioavailability according to seasonal variations. And following another reviewer, we added the references related to the importance of “As bioavailability” for emphasizing our findings. It is below (highlighted red). Again, we have a lacking in this research, not conducting the various experiments for describing the exact leaching mechanisms. So, when we will conduct research on the environmental restoration of this reservoir, we have to contain the various leaching experiment following up your mentions.
“Many studies have investigated quality assessment of many ions to impact on the irrigation water seriously like sodium (Na), potassium (K), calcium (Ca), magnesium (Mg), bicarbonate (HCO3), chloride (Cl) and sulfate (SO4), etc. [14, 15, 16] and heavy metal leaching from various media, including sewage sludge [6, 17, 18], industrial waste [19], coral ash [20, 21, 22], tropical soils [23], and mine tailings [24, 25]. However, the previous research mainly considered with the water body of the stream and river or relative large scale lake and reservoir [22, 23, 26], and the leaching of arsenic from the bottom of an irrigation reservoir has not been a focus of previous studies even though it has significant implications for agriculture. Small agricultural reservoirs have a relatively small storage capacity, and the water level changes frequently depending on the hydrological process with rainfall related watershed runoff and irrigation. So, the sediments of the reservoir are exposed to various environmental conditions, which are exposed to the air during drought period become submerged in water when the high water levels are maintained. Under the these water conditions, as arsenic leaching processes are sensitive to the physical and chemical conditions to which they are exposed, it is important to have a detailed understanding of how the processes respond to various environments. In addition, as various studies stressed and analyzed deposition ([27]) and diffusive gradient ([28, 29, 30, 31]) characteristics related to the arsenic bioavailability (heavy metals and other persistent pollutants) in water and sediments, we considered the seasonal effects (differences of the water quality between wet season and dry season, especially soluble organic matter) on the arsenic leaching characteristics.”
References
DeVries, C. R.; Wang, F. In situ two-dimensional high-resolution profiling of sulfide in sediment interstitial waters. Sci. Technol 2003, 37, 792-797. DOI: 10.1021/es026109j. Davison, W.; Zhang, H. In situ speciation measurements of trace components in natural waters using thin-film gels. Nature 1994, 367, 546-548. Panther, J. G.; Stillwell, K. P.; Powell, K. J.; Downard, A. J. Development and application of the diffusive gradients in thin films technique for the measurement of total dissolved inorganic arsenic in waters. Chim. Acta 2008, 622, 133-142. DOI: 10.1016/j.aca.2008.06.004. Luo, J.; Zhang, H.; Santner, J.; Davison, W. Performance characteristics of diffusive gradients in thin films equipped with a binding gel layer containing precipitated ferrihydrite for measuring arsenic (V), selenium (VI), vanadium (V), and antimony (V). Chem. 2010, 8903-8909. DOI: 10.1021/ac101676w. Bennett, W. W.; Teasdale, P. R.; Panther, J. G.; Welsh, D. T.; Jolley, D. F. New diffusive gradients in a thin film technique for measuring inorganic arsenic and selenium (IV) using a titanium dioxide based adsorbent. Chem. 2010, 82, 7401-7407. DOI: 10.1021/ac101543p.
Comment R.2:
- no taking into account, for potential downstream impacts, of other As carriers - especially solutes or particles "flying" over reservoirs, or sorption/desorption phenomena in the river bed downstream (sediment/water exchange),
Response to Comment R.2: Thank for pointing out the important missing factor in this research. In this study, we wanted to focus on the sedimentation problem in the reservoir due to the continuous inflow of the As-contaminated tailings and sediment from the abandoned gold mine of the upstream, and we analyzed the leaching characteristics of the As-contaminated sediment in the bed of the reservoir according to seasonal variation. In addition, we assumed that the amount of the particles by flying over reservoir from the upstream would be much less than the amount of sediment by inflow and trap efficiency is also very low. However, because we didn’t monitor and analyze the amount and mechanisms exactly, this is lacking in this study. So, when we conducted research on environmental restoration at study area, we have to consider the point what you recommend and will set the plan for analyzing the factor more clearly.
Comment R.3:
- recommendations for reservoir sediment removal do not take into account the possibility of increased impacts through remobilisation - and do not compare with reservoir bed confinement. As expressed in the conclusions, they may be severely misleading for the design of a remediation operation.
Response to Comment R.3: Thank you for your comments on our study. Although we did not include all the results of our research in this manuscript, monitoring results show that even when the arsenic concentration was not detected in a upstream water, high arsenic concentration was observed in the sediment of the water. It shows nearly 50 mg/kg (in sediment). So, we suggested the measures screening the As-contaminated sediment from upper watershed and eliminating the As-contaminated sediment in the reservoir bed.
Comment R.4:
- Line 173: "the concentrations coming from the agricultural land (0.018 mg/L) and the stream bed (0.019 mg/L) were similar to each other" True for stream sediments, but only on average for soils, this hides their very large variability (Figure 3).
Response to Comment R.4: We agree with your point. Following the recommendation, we revealed the large variability of the arsenic concentrations of the samples from the agricultural land and explained the reason (highlighted red), briefly. And finally, we added the new conclusion with this reason. The added contents of conclusion were that “We need to monitor and survey the contamination level from most of the agricultural land and have to restore the contaminated soil, immediately.”
“3.1. Batch-leaching test results
The arsenic concentrations of soils and sediments sampled at three different locations were analyzed using the TCLP test (Figure 4). The highest average arsenic concentration was found in effluent from the reservoir sediment (0.026 mg/L); the average concentrations coming from the agricultural land soil (0.018 mg/L) and the stream sediment (0.019 mg/L) were similar to each other, but the concentrations of the soil samples from the agricultural lands show considerably greater variability (Figure 4). It means that the arsenic concentrations in each agricultural land of this study area are various, and it is considered that all agricultural lands of this study area are not contaminated highly but certain agricultural land which was readjusted using high contaminated soil with mine tailings when the arable lands were readjusted show a high level of arsenic contamination.”
Comment R.5:
- Line 181 "soil from the stream bed" do you mean stream sediment ? or soil formed in the stream bed ?
Response to Comment R.5: We meant the reservoir and the stream sediment which are sampled from the bed and soil from the agricultural land. So, we arranged the terms used in paragraph [3.1. Batch-leaching test results] for showing the meaning what we want to express and corrected [Figure 4] tittle (highlighted in red) for the reason. We also have reviewed the terms (soil, sediment, bed, etc.) used throughout the paper and modified the sentences or expressions for expressing the exact meaning.
“3.1. Batch-leaching test results
The arsenic concentrations of soils and sediments sampled at three different locations were analyzed using the TCLP test (Figure 4). The highest average arsenic concentration was found in effluent from the reservoir sediment (0.026 mg/L); the average concentrations coming from the agricultural land soil (0.018 mg/L) and the stream sediment (0.019 mg/L) were similar to each other, but the concentrations of the soil samples from the agricultural lands show considerably greater variability (Figure 4). It means that the arsenic concentrations in each agricultural land of this study area are various, and it is considered that all agricultural lands of this study area are not contaminated highly but certain agricultural land which was readjusted using high contaminated soil with mine tailings when the arable lands were readjusted show a high level of arsenic contamination.
The contamination potential of soils and sediments is explained by the mobility of pollutants including arsenic. Arsenate (AsO43-) and arsenite (AsO33-), anionic arsenic compounds, form chelates and precipitate in combination with multiple cationic metals. When arsenic adheres to soil and sediment particles with metallic substances (e.g., steel and aluminum), the iron/arsenic (Fe/As) ratios increase and the mobility of arsenate may decrease. In this study, we investigated the Fe/As ratios of the soil and sediment samples as an indicator of their arsenic mobility. The average Fe/As ratio of the reservoir sediment (368) was lower than those from the other soils and sediments (1,819 for the agricultural land soil and 1,914 for the stream sediment), indicating that As in the reservoir sediment was more mobile and had a higher leaching potential than that in other soil and sediment.”
Figure 4. Comparison of arsenic concentrations derived from the batch-leaching test (a: the reservoir sediment, b: the agricultural land soil, c: the stream sediment).
Comment R.6: Some very minor formal corrections: Arsenate (AsO43-) and arsenite (AsO33-) use exponent case for ionic charge
Response to Comment R.6: We made a mistake in editing process, so we corrected the terms expression (highlighted in red) in a right way.
“Arsenate (AsO43-) and arsenite (AsO33-), anionic arsenic compounds, form chelates and precipitate in combination with multiple cationic metals.”

Reviewer 2 Report
The paper faces an actual problem, and with a correct methodology. However, in my modest opinion, the scientific contribution to the issue is very low.
It is possible to arrive at the same conclusions only on the base of the consolidated scientific literature. In this direction the introduction should be improved with references on the in-situ assessment of the arsenic bioavailability (heavy metals and other persistent pollutants) in water and sediments. For instance:
W. Davison, H. Zhang, In situ speciation measurements of trace components in natural waters
using thin-film gels. Nature 1994, 367. 546-548.
C. R. DeVries, F. Wang, In situ two-dimensional high-resolution profiling of sulfide in sediment interstitial waters. Environ. Sci. Technol 2003, 37. 792-797.
J. G. Panther, K. P. Stillwell, K. J. Powell, A. J. Downard, Development and application of the diffusive gradients in thin films technique for the measurement of total dissolved inorganic arsenic in waters. Anal. Chim. Acta 2008, 622. 133-142.
J. Luo, H. Zhang, J. Santner, W. Davison, Performance characteristics of diffusive gradients in thin films equipped with a binding gel layer containing precipitated ferrihydrite for measuring arsenic (V), selenium (VI), vanadium (V), and antimony (V). Anal. Chem. 2010.
W. W. Bennett, P. R. Teasdale, J. G. Panther, D. T. Welsh, D. F. Jolley, New diffusive gradients in a thin film technique for measuring inorganic arsenic and selenium (IV) using a titanium dioxide based adsorbent. Anal. Chem. 2010, 82. 7401-7407
The merit of the authors is to promote a warning message for the local communities. In this direction probably the best conclusion of the work is to explicitly suggest to the farmers that they must test frequently the Arsenic concentration of the reservoir (luckily low cost analytical kits for the self analysis of arsenic exist, and are easy to use).
Below few minor suggestions:
Please, make explicit when the same method was used for both sediments and soils.
Line 93 For the batch-leaching test, the soil samples were collected from the reservoir bed. - For the reservoir bed please use always the same word: sediment.
Lines 104, 105 To prepare a soil-water mixture, 96.5 mL of water was added to a 5 g soil sample (particle sizes less than or equal to 1 mm) in a 500 mL beaker. - Please, describe how the soil samples were prepared for the batch-leaching test. Was the soil air-dried and sieved to 2 mm as for the column-leaching tests?
Lines 151, 152 - The soil test showed that the soil is loamy sand; its arsenic concentration was 56 mg/kg. - This concentration is actually high. For the environmental regulations of most of the Countries this level is high enough to declare the site a polluted site that need recovery (intervention). Please, explain the method adopted for the assessment of Arsenic in soil.
Table 3. Physicochemical properties of soil samples used in column-leaching test. - Please, explain the meaning of all the acronyms (T-P etc.)
Author Response
Responses to Reviewer’s Comments for the Manuscript, applsci-617476
Characteristics of Arsenic Leached from Sediments: Agricultural Implications of Abandoned Mines
Reviewer
Comment R.1: The paper faces an actual problem, and with a correct methodology. However, in my modest opinion, the scientific contribution to the issue is very low.
It is possible to arrive at the same conclusions only on the base of the consolidated scientific literature. In this direction the introduction should be improved with references on the in-situ assessment of the arsenic bioavailability (heavy metals and other persistent pollutants) in water and sediments. For instance:
using thin-film gels. Nature 1994, 367. 546-548. R. DeVries, F. Wang, In situ two-dimensional high-resolution profiling of sulfide in sediment interstitial waters. Environ. Sci. Technol 2003, 37. 792-797. G. Panther, K. P. Stillwell, K. J. Powell, A. J. Downard, Development and application of the diffusive gradients in thin films technique for the measurement of total dissolved inorganic arsenic in waters. Anal. Chim. Acta 2008, 622. 133-142. Luo, H. Zhang, J. Santner, W. Davison, Performance characteristics of diffusive gradients in thin films equipped with a binding gel layer containing precipitated ferrihydrite for measuring arsenic (V), selenium (VI), vanadium (V), and antimony (V). Anal. Chem. 2010. W. Bennett, P. R. Teasdale, J. G. Panther, D. T. Welsh, D. F. Jolley, New diffusive gradients in a thin film technique for measuring inorganic arsenic and selenium (IV) using a titanium dioxide based adsorbent. Anal. Chem. 2010, 82. 7401-7407
Response to Comment R.1: Thank you for your constructive and positive comments on our study and this manuscript. Following your recommendations, we added the references describing the arsenic bioavailability (heavy metals and other persistent pollutants) in water and sediments on the introduction paragraph. The added contents were below (highlighted in red).
“Many studies have investigated quality assessment of many ions to impact on the irrigation water seriously like sodium (Na), potassium (K), calcium (Ca), magnesium (Mg), bicarbonate (HCO3), chloride (Cl) and sulfate (SO4), etc. [14, 15, 16] and heavy metal leaching from various media, including sewage sludge [6, 17, 18], industrial waste [19], coral ash [20, 21, 22], tropical soils [23], and mine tailings [24, 25]. However, the previous research mainly considered with the water body of the stream and river or relative large scale lake and reservoir [22, 23, 26], and the leaching of arsenic from the bottom of an irrigation reservoir has not been a focus of previous studies even though it has significant implications for agriculture. Small agricultural reservoirs have a relatively small storage capacity, and the water level changes frequently depending on the hydrological process with rainfall related watershed runoff and irrigation. So, the sediments of the reservoir are exposed to various environmental conditions, which are exposed to the air during drought period become submerged in water when the high water levels are maintained. Under the these water conditions, as arsenic leaching processes are sensitive to the physical and chemical conditions to which they are exposed, it is important to have a detailed understanding of how the processes respond to various environments. In addition, as various studies stressed and analyzed deposition ([27]) and diffusive gradient ([28, 29, 30, 31]) characteristics related to the arsenic bioavailability (heavy metals and other persistent pollutants) in water and sediments, we considered the seasonal effects (differences of the water quality between wet season and dry season, especially soluble organic matter) on the arsenic leaching characteristics.”
References
[27] DeVries, C. R.; Wang, F. In situ two-dimensional high-resolution profiling of sulfide in sediment interstitial waters. Environ. Sci. Technol 2003, 37, 792-797. DOI: 10.1021/es026109j.
[28] Davison, W.; Zhang, H. In situ speciation measurements of trace components in natural waters
using thin-film gels. Nature 1994, 367, 546-548.
[29] Panther, J. G.; Stillwell, K. P.; Powell, K. J.; Downard, A. J. Development and application of the diffusive gradients in thin films technique for the measurement of total dissolved inorganic arsenic in waters. Anal. Chim. Acta 2008, 622, 133-142. DOI: 10.1016/j.aca.2008.06.004.
[30] Luo, J.; Zhang, H.; Santner, J.; Davison, W. Performance characteristics of diffusive gradients in thin films equipped with a binding gel layer containing precipitated ferrihydrite for measuring arsenic (V), selenium (VI), vanadium (V), and antimony (V). Anal. Chem. 2010, 8903-8909. DOI: 10.1021/ac101676w.
[31] Bennett, W. W.; Teasdale, P. R.; Panther, J. G.; Welsh, D. T.; Jolley, D. F. New diffusive gradients in a thin film technique for measuring inorganic arsenic and selenium (IV) using a titanium dioxide based adsorbent. Anal. Chem. 2010, 82, 7401-7407. DOI: 10.1021/ac101543p.
Comment R.2: The merit of the authors is to promote a warning message for the local communities. In this direction probably the best conclusion of the work is to explicitly suggest to the farmers that they must test frequently the Arsenic concentration of the reservoir (luckily low cost analytical kits for the self analysis of arsenic exist, and are easy to use).
Response to Comment R.2: We agree with your point. Following the recommendation, we revealed the large variability of the arsenic concentrations of the samples from the agricultural land and explained the reason (highlighted red), briefly.
“3.1. Batch-leaching test results
The arsenic concentrations of soils and sediments sampled at three different locations were analyzed using the TCLP test (Figure 4). The highest average arsenic concentration was found in effluent from the reservoir sediment (0.026 mg/L); the average concentrations coming from the agricultural land soil (0.018 mg/L) and the stream sediment (0.019 mg/L) were similar to each other, but the concentrations of the soil samples from the agricultural lands show considerably greater variability (Figure 4). It means that the arsenic concentrations in each agricultural land of this study area are various, and it is considered that all agricultural lands of this study area are not contaminated highly but certain agricultural land which was readjusted using high contaminated soil with mine tailings when the arable lands were readjusted show a high level of arsenic contamination.”
And finally, following your advice, we added the new conclusion with the necessity of the monitoring the water quality of the stream and reservoir and the setting the plan for the agricultural land recovery. The added contents of conclusion were below.
“Based on these study results, for helping farming activities and supplying safe irrigation water from arsenic contamination, local government and irrigation water management agency need to monitor the water quality changes of the reservoir and stream, regularly. In addition, for the ultimate management of the water from arsenic contamination in reservoirs, local government has to set the plan for stopping the inflow of the arsenic-contaminated sediment from the upstream and for the removal of the arsenic-contaminated sediments and soils on the reservoir bed. Another important point is that because this study results also show that there was a large variability of the arsenic leaching concentrations of the samples from each agricultural land, local government and farmers need to survey the contamination level from most of the agricultural land and have to restore the contaminated soil, immediately.”
Comment R.3: Please, make explicit when the same method was used for both sediments and soils.
Response to Comment R.3: In paragraph [2.1. Study area and soil preparation], we added a sentence (highlighted red) for following your comments.
“This study used the following three different tests to investigate the arsenic leaching processes and their responses to the external conditions in detail: the batch- and column-leaching tests and sequential extraction. For the batch-leaching test, the soil samples were collected from the reservoir bed. To identify critical arsenic sources among the various soils, agricultural areas and river beds were also analyzed using the batch-leaching test. Soil samples for the column-leaching tests were prepared by mixing samples taken from six locations on the reservoir bed and then air-dried and sieved to 2 mm. In this study, the same method was applied to soils and sediments for measuring the arsenic concentration.”
Comment R.4: Line 93 For the batch-leaching test, the soil samples were collected from the reservoir bed. - For the reservoir bed please use always the same word: sediment.
Response to Comment R.4: We meant the reservoir and the stream sediment which are sampled from the bed and soil from the agricultural land. So, we have reviewed the terms (soil, sediment, bed, etc.) used throughout the paper and modified the sentences or expressions for showing the exact meaning what we want to express.
Comment R.5: Lines 104, 105 To prepare a soil-water mixture, 96.5 mL of water was added to a 5 g soil sample (particle sizes less than or equal to 1 mm) in a 500 mL beaker. - Please, describe how the soil samples were prepared for the batch-leaching test. Was the soil air-dried and sieved to 2 mm as for the column-leaching tests?
Response to Comment R.5: First of all, I'm sorry for making you misunderstand due to the placement of the sentences. I already wrote it on line 94, 95 following sentence.
“Soil samples for the column-leaching tests were prepared by mixing samples taken from six locations on the reservoir bed and then air-dried and sieved to 2 mm.”
Comment R.6: Lines 151, 152 - The soil test showed that the soil is loamy sand; its arsenic concentration was 56 mg/kg. - This concentration is actually high. For the environmental regulations of most of the Countries this level is high enough to declare the site a polluted site that need recovery (intervention). Please, explain the method adopted for the assessment of Arsenic in soil.
Response to Comment R.6: Because the assessment standard of arsenic contamination in soil is various depending on the law and the characteristics of the soil, environment, the culture and the way of life, etc. on each country, we just considered the act of assessing the level of soil contamination in South Korea. In South Korea, the Minister of Environment (ME) made “SOIL ENVIRONMENT CONSERVATION ACT” to prevent potential hazard to public health and environment to be caused by soil contamination. In soil environment conservation act, ME suggested “Worrisome Level of Soil Contamination” (described in detail below) for 21 pollutant matter by categorizing 3 area group (1st area = residence, 2nd area = agricultural, stream, reservoir, religion, park, etc., 3rd area = military, parking lot, road, etc.). ME made a standard for each 3 area group for pollutant matter in soil, as for arsenic, the standard was set at 25 mg/kg for 1st area, 50 mg/kg for 2nd area, and 200 mg/kg for 3rd area. Because our study area was 2nd area and sampled soil showed average arsenic concentration over than 50 mg/kg (56.84 mg/kg), we regarded our study area as a serious pollution hazard area.
SOIL ENVIRONMENT CONSERVATION ACT
Article 4-2 (Worrisome Level of Soil Contamination)
The level of soil contamination, which is likely to obstruct the health and properties of persons or rearing of animals and plants (hereinafter referred to as the "worrisome level") shall be prescribed by Ordinance of the Ministry of Environment.
[This Article Wholly Amended by Act No. 10551, Apr. 5, 2011]
Comment R.7: Table 3. Physicochemical properties of soil samples used in column-leaching test. - Please, explain the meaning of all the acronyms (T-P etc.)
Response to Comment R.7: We already expressed the whole words in the main paragraph, but for helping readers understand the meaning from the table, we modified the contents (highlighted in red) of the table following the suggestions.
Table 3. Physicochemical properties of soil samples used in column-leaching test.
pH |
Organic matter (%) |
CEC (cmol/kg) |
Total Nitrogen (TN) (mg/kg) |
Total phosphorus (TP) (mg/kg) |
Available phosphorus (mg/kg) |
As (mg/kg) |
Texture |
6.1 |
2.68 |
6.19 |
1,120 |
374.2 |
21.9 |
56.34 |
Loamy sand |

Reviewer 3 Report
The article entitled "Characteristics of Arsenic Leached from Sediments: Agricultural Implications of Abandoned Mines" written by Soonho Hwang et al. is interesting and reports the impact of arsenic in the agricultural field. The paper is well structured and written. The paper just needs a very brief minor revision based on the following comments below:
In the section introduction, page 2, line 47. The authors should include not just the transport of heavy metals but the rest of ions due to agricultural overexplotation etc. There are more elements that have a serious impact on the irrigation water, including the agricultural activity. I recommend to include some recent papers about it (i.e. Groundwater Quality Assessment in a Volcanic Mountain Range, etc.). It would be good as most of cited papers are little bit old. In section results, page 6, lines 175 and 176, please, write 3- as superscript for the arsenate and arsenite respectively.Author Response
Responses to Reviewer’s Comments for the Manuscript, applsci-617476
Characteristics of Arsenic Leached from Sediments: Agricultural Implications of Abandoned Mines
Reviewer
Comment R.1: In the section introduction, page 2, line 47. The authors should include not just the transport of heavy metals but the rest of ions due to agricultural overexplotation etc. There are more elements that have a serious impact on the irrigation water, including the agricultural activity. I recommend to include some recent papers about it (i.e. Groundwater Quality Assessment in a Volcanic Mountain Range, etc.). It would be good as most of cited papers are little bit old.
Response to Comment R.1: Thank you for your constructive and positive comments on our study and this manuscript. Following the suggestions, we searched for more research and paper relating to other ions influencing on the irrigation water quality. And we modified our sentence (highlighted in red) and added the latest three papers in the references.
“Many studies have investigated quality assessment of many ions to impact on the irrigation water seriously like sodium (Na), potassium (K), calcium (Ca), magnesium (Mg), bicarbonate (HCO3), chloride (Cl) and sulfate (SO4), etc. [14, 15, 16] and heavy metal leaching from various media, including sewage sludge [6, 17, 18], industrial waste [19], coral ash [20, 21, 22], tropical soils [23], and mine tailings [24, 25].”
References
Ruiz-García, A.; Carrascosa-Chisvert, M.D.; Mena, V.; Souto, R.M.; Santana, J.J.; Nuez, I. Groundwater Quality Assessment in a Volcanic Mountain Range (South of Gran Canaria Island, Spain). Water 2019, 11, 754. DOI: 10.3390/w11040754. Chen, J.; Huang, Q.; Lin, Y.; Fang, Y.; Qian, H.; Liu, R.; Ma, H. Hydrogeochemical Characteristics and Quality Assessment of Groundwater in an Irrigated Region, Northwest China. Water 2019, 11, 96. DOI: 10.3390/w11010096. Sridharan, M.; Senthil Nathan, D. Groundwater quality assessment for domestic and agriculture purposes in Puducherry region. Applied Water Science 2017, 7(7), 4037-4053. DOI: 10.1007/s13201-017-0556-y.Comment R.2: In section results, page 6, lines 175 and 176, please, write 3- as superscript for the arsenate and arsenite respectively.
Response to Comment R.2: We made a mistake in editing process, so we corrected the terms expression (highlighted in red) in a right way.
Arsenate (AsO43-) and arsenite (AsO33-), anionic arsenic compounds, form chelates and precipitate in combination with multiple cationic metals.
